# The Potential Role of *Nigella sativa* Seed Oil as Epigenetic Therapy of Cancer

**DOI:** 10.3390/molecules27092779

**Published:** 2022-04-27

**Authors:** Safialdin Alsanosi, Ryan A. Sheikh, Sultan Sonbul, Hisham N. Altayb, Afnan S. Batubara, Salman Hosawi, Kaltoom Al-Sakkaf, Omeima Abdullah, Ziad Omran, Mahmoud Alhosin

**Affiliations:** 1Biochemistry Department, Faculty of Science, King Abdulaziz University, Jeddah 21589, Saudi Arabia; salsanosi0004@stu.kau.edu.sa (S.A.); rsheikh@kau.edu.sa (R.A.S.); ssonbul@kau.edu.sa (S.S.); hdemmahom@kau.edu.sa (H.N.A.); shosawi@kau.edu.sa (S.H.); 2Centre for Artificial Intelligence in Precision Medicines, King Abdulaziz University, Jeddah 21589, Saudi Arabia; 3College of Pharmacy, Umm Al-Qura University, Makkah 21955, Saudi Arabia; asbatubara@uqu.edu.sa (A.S.B.); oaabdullah@uqu.edu.sa (O.A.); 4Medical Laboratory Technology Department, Faculty of Applied Medical Sciences, King Abdulaziz University, Jeddah 21589, Saudi Arabia; kalsakkaf@kau.edu.sa; 5Department of Pharmaceutical Sciences, Pharmacy Program, Batterjee Medical College, Jeddah 21442, Saudi Arabia; ziad.omran@bmc.edu.sa

**Keywords:** *Nigella sativa* oil, thymoquinone, epigenetic, cancer, UHRF1, DNMT1, HDAC1

## Abstract

*Nigella sativa* oil, commonly known as black seed oil (BSO), is a well-known Mediterranean food, and its consumption is associated with beneficial effects on human health. A large number of BSO’s therapeutic properties is attributed to its pharmacologically active compound, thymoquinone (TQ), which inhibits cell proliferation and induces apoptosis by targeting several epigenetic players, including the ubiquitin-like, containing plant homeodomain (PHD) and an interesting new gene, RING finger domains 1 (UHRF1), and its partners, DNA methyltransferase 1 (DNMT1) and histone deacetylase 1 (HDAC1). This study was designed to compare the effects of locally sourced BSO with those of pure TQ on the expression of the epigenetic complex UHRF1/DNMT1/HDAC1 and the related events in several cancer cells. The gas chromatographs obtained from GC-MS analyses of extracted BSO showed that TQ was the major volatile compound. BSO significantly inhibited the proliferation of *MCF-7*, HeLa and Jurkat cells in a dose-dependent manner, and it induced apoptosis in these cell lines. BSO-induced inhibitory effects were associated with a significant decrease in mRNA expression of UHRF1, DNMT1 and HDAC1. Molecular docking and MD simulation showed that TQ had good binding affinity to UHRF1 and HDAC1. Of note, TQ formed a stable metal coordinate bond with zinc tom, found in the active site of the HDAC1 protein. These findings suggest that the use of TQ-rich BSO represents a promising strategy for epigenetic therapy for both solid and blood tumors through direct targeting of the trimeric epigenetic complex UHRF1/DNMT1/ HDAC1.

## 1. Introduction

Cancer is a complex disease that results from various factors, such as genetic mutations, but also from epigenetic alterations including abnormal DNA methylation and histone deacetylation [1,2,3,4]. These epigenetic changes are involved in the silencing of various tumor suppressor genes (TSGs) in cancer. Maintenance of DNA methylation patterns is indispensable for cell proliferation. During cell division, the duplication of symmetric DNA methylation is ensured by the well-documented interaction between the ubiquitin-like, containing PHD and RING domains (UHRF1) and the DNA methyltransferase 1 (DNMT1) in tandem [5,6,7,8,9,10]. Indeed, the SRA (SET and RING-associated) domain of UHRF1 recognizes hemi-methylated DNA generated during DNA replication which allow UHRF1 to recruit DNMT1 in order to methylate the newly synthesized DNA strand, resulting in silencing of many TSGs in cancer [5,6,7,8,9,10,11,12,13,14]. UHRF1 can also regulate the status of histone acetylation through a direct interaction with the histone deacetylase 1 (HDAC1) [15,16,17]. Thus, targeting the UHRF1-mediated epigenetic machinery is considered a promising approach to change the altered epigenetic state in cancer cells and allowing the re-expression of TSGs. Many plant-derived components are known for their bioactivities that support prevention and treatment of several diseases, including cancer [18,19,20]. In recent years, this approach has gained more popularity as an alternative to chemical drugs. Interestingly, combining synthetic drugs with natural compounds could increase the anticancer activity and decrease severe side effects of such treatment [21,22,23].

Black seed oil (BSO), also known as *Nigella sativa* oil, is an important source of components with a wide range of biological activities, such as cytotoxic effects on a panel of cancer cell lines including colon cancer, lung cancer and human breast cancer cells [24,25,26,27]. Besides its anticancer effects against many types of tumors, black seeds and their oil have remedial usage for other various diseases and disorders such as diabetes, cardiovascular disease, hypertension, infection, inflammation and eczema [24,25]. Several studies reported that the pharmacological activities and therapeutic benefits of BSO are mainly attributed to thymoquinone (TQ) [25,27,28,29,30] which can inhibit cell proliferation and induce apoptosis through the targeting of several epigenetic regulators, including the trimeric complex UHRF1/DNMT1/HDAC1 [31,32,33,34,35,36]. Thus, the food value of BSO is not related only to its safety for human and medicinal uses [30,37,38,39], but also to its considerable content of thymoquinone [40,41].

The aim of the present study was to extract BSO from black seeds sourced from the local market of Al-Qassim, Saudi Arabia, to determine its content of TQ and to investigate its inhibitory effects on the expression of UHRF1, DNMT1 and HDAC1 and the related events in several cancer cells.

## 2. Results

### 2.1. Qualitative Analysis of N. sativa Seed Compounds

Several reports have shown that N. sativa seeds are rich in volatile constituents, but the compositions are different according to geographical origin of the plants [40,42,43,44]. A recent report showed that BSO content of TQ can vary from as low as 0.01 mg/g to 13.30 mg/g [45]. The gas chromatographs obtained from GC-MS analyses of oil extracted using SFE showed that thymoquinone and thymohydroquinone were the major volatile compounds in the extracted BSO (Figure 1). Linoleic acid, oleic acid and petroselinic acid were found to make up the main fatty acid composition identified in the extracted BSO (Figure 1).

### 2.2. Quantitative Analysis of Thymoquinone in the Extracted BSO

Standard stock and working standard solutions of thymoquinone were prepared as shown in “Materials and Methods”. An amount of 500 μL of BSO was diluted in 20 mL of methanol. An HPLC analysis performed on a C18 column using water and methanol as the mobile phase showed that the extracted BSO contained 5.9% TQ (Figure 2). 

### 2.3. The Cytotoxicity Effect of BSO on Cancer Cells

The effect of either BSO or pure TQ was then evaluated on cell viability of different cancer cell lines using WST-1 staining for 24 h of treatment (Figure 3). BSO showed a significant decrease in the cell viability to 84.33% at a concentration of 0.6% (*V*/*V*) and 75.00% at a concentration of 1.25% (*V*/*V*) in JK cells (Figure 3A). MCF7 cell viability was significantly reduced to 90.67% and 82.6% using 0.6 and 1.25% (*V*/*V*) of BSO, respectively (Figure 3B). Under similar experimental conditions, there was a significant decrease in HeLa cell viability to 94% and 85% using 0.6 and 1.25% (*V*/*V*) of BSO, respectively (Figure 3C). The same observations were found when cancer cells were treated with pure TQ at a concentration of 30 µM for Jurkat cells (Figure 3A) and MCF7 cells (Figure 3B) and at 200 µM for HeLa cells (Figure 3C). TQ treatment significantly decreased cell viability of these cell lines, indicating that the anti-proliferative of BSO on cancer cells can be attributed to its main compound, TQ.

### 2.4. The Pro-Apoptotic Effect of BSO on Cancer Cells

To investigate whether BSO-induced proliferation inhibition in cancer is related to the induction of the apoptosis process, the rate of apoptotic cells was analyzed using flow cytometry. Cancer cells were treated with 0.6 and 1.25% (*V*/*V*) of BSO for 24 h and apoptosis was then assessed as described in “Materials and Methods”. Increasing concentrations of BSO induced an increasing number of apoptotic cells in JK cells (Figure 4A), MCF7 cells (Figure 4B) and HeLa cells (Figure 4C). BSO at 1.25% (*V*/*V*) significantly increased the apoptotic rate to 33.57% in Jurkat cells, (Figure 4A), to 21.8% in MCF7 cells (Figure 4B) and to 19.1% in HeLa cells (Figure 4B). As expected, 30 µM of pure TQ increased the apoptosis rate in Jurkat cells to 80% (Figure 4A) and in MCF7 cells to 30% (Figure 4B), whereas this percentage in HeLa cells reached 30.1% using 200 µM of TQ (Figure 4C). Marked changes in morphological characteristics were observed after 24 h of BSO exposure (Figure 5). Both BSO and TQ treatments led to JK cells shrinking (Figure 5A) and increased the number of detached MCF7 cells (Figure 5B) and HeLa cells (Figure 5C) compared to untreated cells. These BSO-induced morphological changes of cancer cells are an indication of the typical appearance of apoptotic cells. Together, these findings suggest that BSO-induced apoptosis in cancer cells and this effect could be related to the pro-apoptotic activity of its main compound TQ.

### 2.5. The Effect of BSO on the mRNA Expression of UHRF1, DNMT1 and HDAC1 in Cancer Cells

Since both DNMT1 and HDAC1 are well-documented partners of the epigenetic reader UHRF1 in cancer cells [31], we studied the 24 h effect of BSO on the mRNA expression of the trimeric complex UHRF1, DNMT1 and HDAC1 in cancer cells. We found that the mRNA expression of the target genes was significantly decreased in a dose-dependent manner in JK cells (Figure 6A), MCF7 cells (Figure 6B) and HeLa cells (Figure 6C) treated with either BSO or with pure TQ compared to the control. This suggests that these epigenetic regulators have a significant role in the anti-proliferative and pro-apoptotic effects of BSO in cancer cells and that its main component TQ could be the main regulator of these enzymes.

### 2.6. Molecular Docking and MD Simulation 

To investigate whether TQ can interact with UHRF1 and/or its partners, DNMT1 and HDAC1, we analyzed the molecular interactions of the TQ-protein as described in “Materials and Methods”. The interaction between TQ and UHRF1 generated a binding energy of −6.5 kcal/mol and one hydrogen bond with ASN194 (Figure 7A1,A2, Table 1). MD simulation was used to study the conformational changes of ligands on protein backbones and to examine the complex stability. The RMSD of the TQ ligand on the UHRF1 backbones showed a small deviation before 15 ns of simulation, then remained aligned with protein backbones until the end; as a result of this interaction one stable hydrogen bond (existed at more than 70%) was generated with HIS185, and four hydrophobic interactions were formed with PHE152, VAL157, TYR191 and VAL196 (Figure 8A2). Also, TQ exhibited −5 kcal/mol docking energy and two hydrogen bonds with LYS697 and LYS1242 of human DNMT1 (Figure 7B, Table 1). As shown in Figure 8B1, TQ is deviated from DNMT1 protein backbones during the simulation and remained separated until the end of the simulation, and most of the bonds formed were non-stable as they existed for less than 40% of the simulation time. Excellent docking energy (−8.3 kcal/mol) was generated from the interaction of the ligand with HDAC1 protein (Table 1), this favorable energy was the result of the interaction of TQ and zinc atom with active site residues of the target protein. Zinc atom formed a strong metal coordinate bond with a small distance (2.2Å) (Figure 7C2. Moreover, TQ showed excellent stability in the backbones of HDAC1, which remained aligned with the protein from the start of the simulation until the end (Figure 8C1). Also, three stable ionic bonds between protein and zinc ions were observed for 100% of the simulation time (Figure 8C2). This stability of the TQ and HDAC1 complex is reflected in the low number (20 Å) of SASA generated.

## 3. Discussion

*Nigella sativa*, classified under the Ranunculaceae family [30], has a wide spectrum of potential pharmacological uses due to its anticancer properties. Several in vitro, in vivo and clinical studies have highlighted the promising anticancer activities of *N*. *sativa* and its main active compound TQ [25,46,47,48]. Different cell toxicity studies have reported that *N. sativa* oil and TQ are safe for human use and consumption [30,37,38,39]. 

Thymoquinone determination in the extracted BSO is one of the main goals of the present study. The gas chromatographs obtained from GC-MS analyses of extracted oil showed that thymoquinone was the major volatile compound in the extracted BSO. An HPLC analysis revealed that BSO contained 5.9% of TQ. These findings are in line with the results of previous studies that pointed out that TQ is one of the major volatile compounds in the N. sativa essential oil [40,41].

Several in vitro and in vivo studies have shown that *N. sativa* volatile oil exerts anti-cancer activity by targeting several signaling pathways, especially those involved in cell proliferation and apoptosis [21,49,50,51]; however, to our knowledge this is the first study that evaluates the inhibitory effect of *N. sativa* volatile oil on the epigenetic code of cancer cells regulated by the trimeric complex UHRF1/DNMT1/HDAC1. The integrated epigenetic UHRF1 is overexpressed in many solid and hematological tumors and its overexpression is considered as a main cause of enhanced cell proliferation and defective apoptosis through the inhabitation of several tumor suppressor genes [52,53,54,55]. Several studies have shown that TQ can decrease UHRF1 expression in cancer cells with the subsequent induction of apoptosis [31,32,33,34,35,36]. The present study showed that BSO decreased the expression of UHRF1 in a panel of cancer cells. BSO-induced UHRF1’s downregulation in cancer cells could be in large part attributed to the high content of thymoquinone in BSO. This hypothesis is supported by the fact that TQ can specifically target the really interesting new domain (RING domain) of UHRF1, leading to a fast auto-polyubiquitination of UHRF1 in cancer cells, which was found to be a prerequisite for UHRF1 degradation in response to TQ that leads to induced apoptosis [32,56]. Furthermore, a molecular docking study revealed that TQ had favorable interactions with the SRA domain UHRF1 [57]. Through its SRA domain, UHRF1 can interact with DNMT1 during DNA replication [7,10]. Moreover, TQ was shown to interact with the catalytic pocket of DNMT1 and to exert in vitro cytotoxicity effects against leukemia by inhibiting the activity of DNMT1, inducing global DNA hypomethylation [58]. In the same context, TQ decreased the expression of DNMT1 and HDAC1 in JK cells and this effect was associated with cell proliferation inhibition and apoptosis induction [34,35,36]. The present study showed that BSO downregulated in JK cells, as an experimental model of blood, and in HeLa and MCF7 cells, as examples of solid malignancies, had the expression of both DNMT1and HDAC1, which are well known partners of UHRF1 [31], suggesting that the use of TQ-rich BSO represents a promising strategy for epigenetic therapy for both solid and blood tumors through direct targeting of UHRF1 and/or its partners DNMT1 and HDAC1. This is supported in the present study by the fact that TQ formed four hydrophobic interactions with PHE152, VAL157, TYR191 and VAL196, all of which are located in the tandem Tudor domain (TTD) of UHRF1, a domain responsible for specific and strong binding to histone H3 di- or trimethylated at lysine 9 [12]. Additionally, TQ bound to human DNMT1 and interacted with LYS1242, which is located in the methyltransferase domain of DNMT1 [59]. The present study also showed that docking and MD simulation of TQ to the active site of the HDAC1 had a low binding affinity (−8.3 kcal/mol), SASA value of 20 Å and a high MolSA of 184Å. Usually, higher values of PSA and MolSA and lower RMSD and SASA values in the system indicate better stability of the complex [60,61,62]. Interestingly, TQ showed a stable interaction with the zinc ion that was found in the active site of the HDAC1, which played a crucial role in maintaining the stability of the protein catalytic site [63], indicating that TQ is a specific regulator of HDAC1 in cancer. Recently, TQ-rich BSO was shown to regulate the expression of HDAC1 through an in vitro model of low-grade inflammation of human macrophages [64].

## 4. Materials and Methods

### 4.1. Chemicals and Reagents

Black seeds were obtained from the local market of Al-Qassim, Saudi Arabia. The standard TQ was purchased from Sigma–Aldrich (St. Louis, MO, USA). HPLC-grade methanol (Honeywell, Seelze, Germany) and HPLC-grade water (Sigma Aldrich, Basel, Switzerland) were used for the preparation of the HPLC mobile phase.

### 4.2. Supercritical Fluid Extraction System (SCF)

The extraction was performed using an SFT-110 SFE system (Supercritical Fluid Technologies, Inc. Newark, DE, USA) according to the manufacturer’s manual. An amount of 20 g of black seeds was placed in a 25 mL supercritical fluid extractor at its full capacity and was extracted using supercritical carbon dioxide. The temperature and pressure were set at 25 °C and 180 bar, respectively. After the static period (60 min), the extract-laden supercritical CO_2_ (constant flow rate 2.0 mL min^−1^) was sent to the extractor vessel. The extracted oil from the black seeds was then collected, at a medium pressure of 4000 psi and room temperature (25 °C), in test tubes for 15 min and the SFE-CO_2_ extract was stored for future use.

### 4.3. Determination of Bioactive Compounds in the Extracted Oil

The constituents of the SFE-CO2 extract of black seeds were analyzed using the Agilent 7890B GC instrument connected to the EI triple quadruple mass spectrometer, 7000C (Agilent technologies, Santa Clara, CA, USA). The chromatographic separation of the fatty acid methyl esters (FAMEs) was performed using the capillary column DB-5; 30 m long, 250 μm Id and 0.2 μm film thickness (catalogue # 122-5032, Agilent technologies, USA). The stock solution of SFE-CO2 extract was analyzed by directly injecting 1 µL into a multi-mode inlet (MMI) operated in splitless mode. The following GC settings were used; the initial oven temperature was 80 °C, ramped at a rate of 10 °C min^−1^ to 150 °C with a 0 min hold time and then ramped at a rate of 5 °C min^−1^ to 250 °C and held for 0 min. Finally, the oven temperature was ramped at a rate of 20 °C/min to 270 °C and was held for 6 min; therefore, the total run time was 34 min. Furthermore, the following parameters were used: injection temperature, 270 °C; auxiliary temperature, 320 °C; helium quench gas flow rate, 4 mLmin^−1^; nitrogen collision gas flow rate, 1.5 mL min^−1^ and helium (carrier gas) flow rate, 1.0 mL min^−1^. The triple quadrupole was auto tuned according to the manufacturer tune manual, with a scan rate set from 35 to 700 Da and a dwell time of 100 msec. Identification of separated compounds was performed by the National Institute of Standards and Technology Library’s 14.L mass spectra database. The integration of the chromatographic peaks and quantification was performed using MassHunter Version B08.00 qualification and quantification software (Agilent technologies, USA).

### 4.4. HPLC Detection and Quantification of Thymoquinone

Detection and quantification of thymoquinone was carried out on the HPLC 1260 Infinity II LC system, composed of the following modules: a 1260 Quat Pump, 1260 Vialsampler equipped with cooling, 1260 MCT column thermostat and 1260 diode array detector (Agilent Technologies, Waldbronn, Germany). Separation was performed using a Poroshell 120 Eclipse-C18 column (dimensions 46 mm × 100 mm, 2.7 μm) with a flow rate of 2 mL/min and 10 µL injection volume. A mobile phase ratio of water-methanol (50:50% *V*/*V*) in isocratic mode was used. For the preparation of the calibration curve, a standard stock solution of thymoquinone was prepared by dissolving 1mg of thymoquinone in 5 mL of methanol. The working standard solution of thymoquinone (20, 40, 60, 80, 100 and 200 ppm) was prepared by diluting the standard stock solution with the mobile phase. An amount of 500 μL of BSO was diluted in 20 mL of methanol. Prior to HPLC analysis, all the solutions prepared were filtered through a 0.45 µm syringe filter. TQ was detected at 254 nm. Data acquisition, analysis and reporting were performed using Open lab CDS-ChemStation Rev. C. 01.09 software (Agilent Technologies, Waldbronn, Germany, 2018). 

### 4.5. Cell Culture and Treatment 

Jurkat (human T lymphocyte), MCF-7 (human breast cancer) and HeLa (human cervical cancer) cell lines were purchased from the American Type Culture Collection (Manassas, VA, USA) and maintained in a humidified incubator at 37 °C in 5% CO_2_. Jurkat cells were grown in an RPMI 1640 culture medium (Sigma-Aldrich, St-Louis, MO, USA), whereas *MCF-7* and HeLa cells were grown in Dulbecco’s modified Eagle medium (DMEM) (Sigma-Aldrich, St-Louis, MO, USA). All media were supplemented with 15% (*V*/*V*) fetal calf serum (FCS; Biowhittaker, Lonza, Belgium), 2 mM glutamine, 100 U/mL penicillin and 50 μg/mL streptomycin (Sigma, St. Louis, MO, USA). BSO was prepared as 0.6 and 1.25 (*V*/*V*) in 10% DMSO, whereas TQ was prepared as a stock solution of 10 mM. The required working concentrations were prepared by dilution with the cell culture medium. The final concentration of DMSO was always kept less than 0.1% in both control and treated conditions. 

### 4.6. Cell Proliferation Assay

The effect of the extracted BSO and TQ on cell proliferation was analyzed by a colorimetric cell proliferation assay using the WST-1 Cell Proliferation Reagent Kit (Sigma-Aldrich, USA). Briefly, the cells were seeded into 96-multiwell plates at a density of 4 × 10^4^/well for Jurkat cells and 10^4^/well for HeLa and *MCF-7* cells. After 24 h of incubation, the cells were treated with different concentrations of either BSO (*V*/*V*) or TQ for 24 h. The cell proliferation rate was then evaluated by a rapid WST-1 reagent. After incubation at different time intervals, 10 µL of the WST-1 solution was added and the mixture was incubated for a further 3 h at 37 °C. Finally, the absorbance was measured at 450 nm using a microplate ELISA reader (ELx800™ Biotek, Winooski, VT, USA) and the results were analyzed using the Gen5 software (Biotek, USA). The percentage of cell viability was calculated by assuming that the untreated control samples were 100% viable.

### 4.7. Morphological Assay

Cancer cells were seeded in 6-multi-well plates at a density of 5 × 10^5^ cells/well for JK cells and 2 × 10^5^ cells/well for *MCF-7* and HeLa cells, grown for 24 h and followed up with treatment of BSO and TQ for 24 h. The cells’ morphological changes were observed and captured using a phase-contrast microscope (20×) (Nikon, NY, USA).

### 4.8. Apoptosis Assay

Jurkat cells at a concentration of 4 × 10^4^ cells/well in a 96-well plate and *MCF-7* and HeLa cells at a concentration of 2 × 10^5^ cells/well in a 6-well plate were cultured overnight and treated with different concentrations of BSO and TQ for 24 h. The rate of cell apoptosis was analyzed by the Annexin V Binding Guava Nexin^®^ Assay using capillary cytometry (Guava Easycyte Plus HP system, with absolute cell count and six parameters) according to the manufacturer’s recommendations (Guava Technologies Inc., Hayward, CA, USA). Briefly, 100 µL of the nexin reagent staining solution (Millipore^®^, Billerica, MA, USA, catalog no. 4500-0450), containing annexin V-fluorescein and 7AAD, was added and incubated in the dark for 20 min at room temperature. The forward and side scatter were recorded at 10,000 events. The subsequent percentages of the early and the late apoptotic cells were analyzed using the Guava^®^ easyCyte 12HT Benchtop Flow Cytometer (Millipore^®^, Billerica, MA, USA). The results were plotted using the InCyte™ software (Millipore^®^, Billerica, MA, USA).

### 4.9. Reverse Transcription and Real-Time PCR

Cells were treated with 0.6 and 1.25 of BSO (*V*/*V*) and different concentrations of TQ for 24 h. The total RNA was isolated and purified from cancer cells using the RNeasy kit (Haven Scientific, Jeddah, Makkah, Saudi Arabia). The cDNA libraries were created from the RNA (Superscript III Reverse Transcriptase, Invitrogen) by using specific primers and real-time PCR was performed using the SYBR Green qPCR (iQ SUPERMIX, BioRad) on the ABI7500 system. The qPCR conditions were maintained at 95 °C, 30 s; 60 °C, 40 s and 72 °C, 40 s. The results were normalized to those obtained with GAPDH mRNA. The sequences of the primers used for the PCR amplification were: UHRF1 (sense: 5′-GTCGAATCATCTTCGTGGAC-3′; antisense: 5′AGTACCACCTCGCTGGCA-3′); DNMT1 (sense: 5′-GGCCTTTTCACCTCCATCAA-3′; antisense: 5′-GCACAAACTGACCTGCTTCA-3′); HDAC1 (sense: 5′-GCTTGCTGTACTCCGACATG-3′; antisense: 5′-GACAAGGCCACCCAATGAAG-3′) and GAPDH (sense: 5′- GGTGAAGGTCGGA-GTCAAC-3′; antisense: 5′-AGAGTTAAAAGC-AGCCCTGGTG-3′). Amplicons were size-controlled on agarose gel and purity was assessed by analysis of the melting curves at the end of the RT-PCR reaction.

### 4.10. Molecular Docking

Protein and compound preparation of the crystal structures of UHRF1 (PDB ID: 5XPI), DNMT1 (PDB ID: 3PTA) and HDAC1 (PDB ID: 4BKX) was achieved by the Maestro interface’s Prime tool (Schrödinger-Release-2020-3, LLC, New York, NY, USA, 2020). The crystal structures were used as targets for molecular docking and molecular simulation studies, and were downloaded from the RCSB PDB database [65]. The structure of the thymoquinone compound (Compound CID: 10281) was obtained from the PubChem database [66]. The Protein Prep Wizard tool in Maestro was used for the preparation of the proteins’ 3D structures; an ionization state was generated at pH 7.4, there was the addition of hydrogen bonds, and the overall 3D structures’ energy was minimized. The Receptor Grid Generation Maestro interface was used for the prediction of the active site. The binding affinity of thymoquinone to the protein active site was evaluated using the induced fit docking of the Maestro interface. 

### 4.11. Molecular Dynamic (MD) Simulation

The stability of the generated complex from the docking study was evaluated by MD simulation, using the Desmond package in the Schrödinger Maestro software (Release, 2020). A box measuring 10Å was filled using the TIP4P water model for the solvation step and OPLS3e was used for system energy minimization. After that, the ions in the system were nebulized. The complex was equilibrated at 300 K and ran for 50 ns at a pressure (bar) of 1.01325. Additionally, the Desmond parameters of the radius of gyration (rGyr), solvent-accessible surface area (SASA), polar surface area (PSA) and molecular surface area (MolSA) (equivalent to a van der Waals surface area) were used for the estimation of complex stability. 

### 4.12. Statistical Analysis

All the data were presented as the mean ± SD of triplicates performed in the same experiment or an average of at least three separate experiments. Statistical analysis was performed using a one-way ANOVA followed by Tukey’s post hoc test using GraphPad Prism 6 (Graph Pad Software, San Diego, CA, USA) and the significant differences were indicated as * *p* < 0.05, ** *p* < 0.01, *** *p* < 0.001 and **** *p* < 0.0001.

## 5. Conclusions

The present study shows that TQ-rich BSO inhibited cell proliferation and induced apoptosis in cancer cells. The underlying mechanism could involve the interaction of TQ and downregulation of the epigenetic regulators UHRF1, DNMT1 and HDAC1, which normally would promote cell proliferation and protect cells from apoptosis by inhibiting the expression of various tumor suppressor genes. However, the detailed mechanisms of BSO’s effects on the epigenetic code of cancer cells still require further investigation. Objectively, due to the existence of other active compounds in the extracted BSO which also can induce epigenetic alterations, further studies are needed to explore the mechanisms by which BSO regulates the epigenome of cancer cells and how the UHRF1/DNMT1/HDAC1 complex could be involved in this process. These findings could be an additional useful step towards further studies on *N. sativa* and cancer and including BSO and its main component TQ as a regular intake in the dietary treatments that support traditional therapies to achieve the best therapeutic plans.

## Figures and Tables

**Figure 1 molecules-27-02779-f001:**
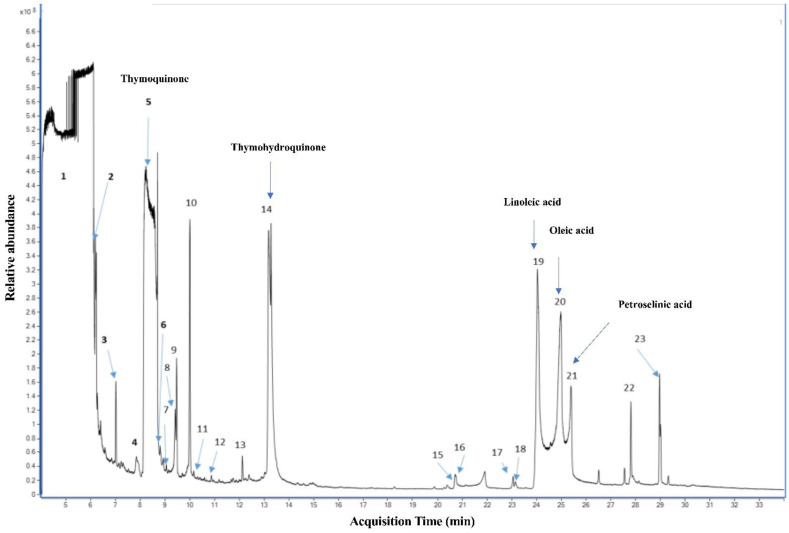
Qualitative GC-MS analyses of N. sativa seed compounds.

**Figure 2 molecules-27-02779-f002:**
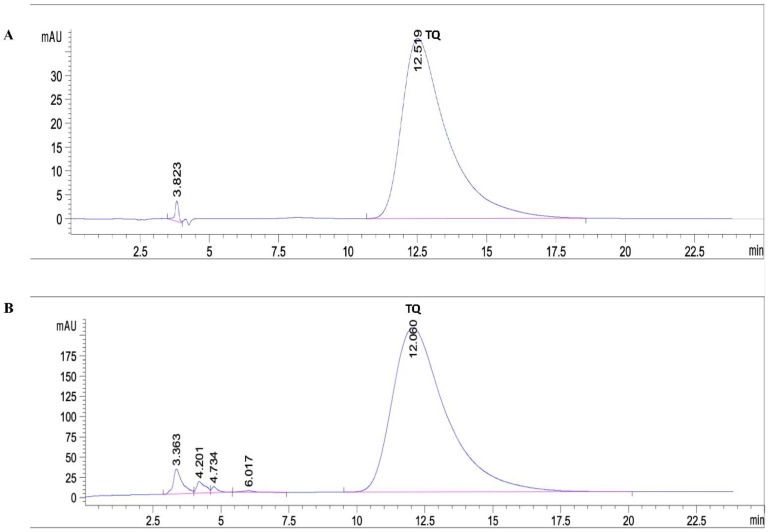
Quantitative HLPC analysis of TQ in the extracted BSO. HPLC analysis for standard solutions of TQ (**A**) and extracted BSO (**B**).

**Figure 3 molecules-27-02779-f003:**
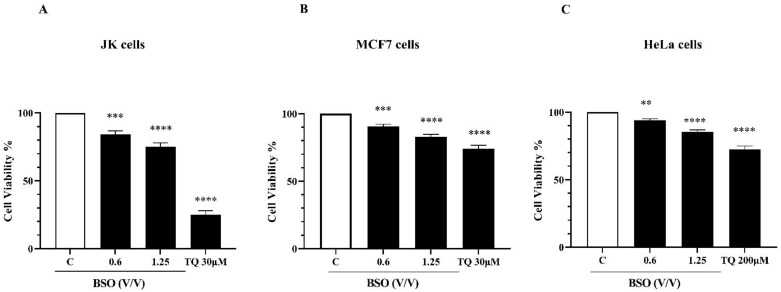
Effect of BSO and TQ on cell viability in cancer cells. Jurkat cells (**A**), MCF7 cells (**B**) and HeLa cells (**C**) were exposed to increasing concentrations of either BSO or TQ for 24 h. Cell viability rate was assessed by WST-1 assay. Values are shown as means ± SD (n = 3); *** p* < 0.01, **** p* < 0.001 and ***** p* < 0.0001 versus respective control.

**Figure 4 molecules-27-02779-f004:**
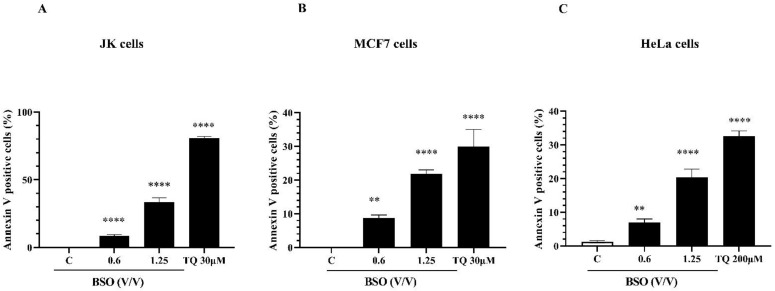
Effect of BSO and TQ on apoptosis in cancer cells. Jurkat cells (**A**), MCF7 cells (**B**) and HeLa cells (**C**) were exposed to increasing concentrations of either BSO or TQ for 24 h. Apoptosis was assessed by flow cytometry using the Annexin V/7AAD staining apoptosis assay. Values are shown as means ± SD (n = 3); *** p* < 0.01, and ***** p* < 0.0001 versus respective control.

**Figure 5 molecules-27-02779-f005:**
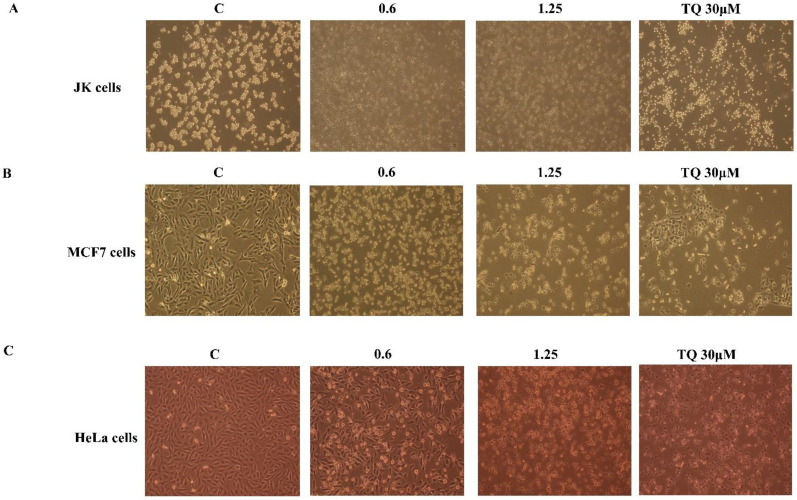
Morphological changes of cancer cells in response to BSO and TQ. Jurkat cells (**A**), MCF7 cells (**B**) and HeLa cells (**C**) were exposed to increasing concentrations of either BSO or TQ for 24 h. The cells’ morphological changes were observed and captured using a phase-contrast microscope (20×).

**Figure 6 molecules-27-02779-f006:**
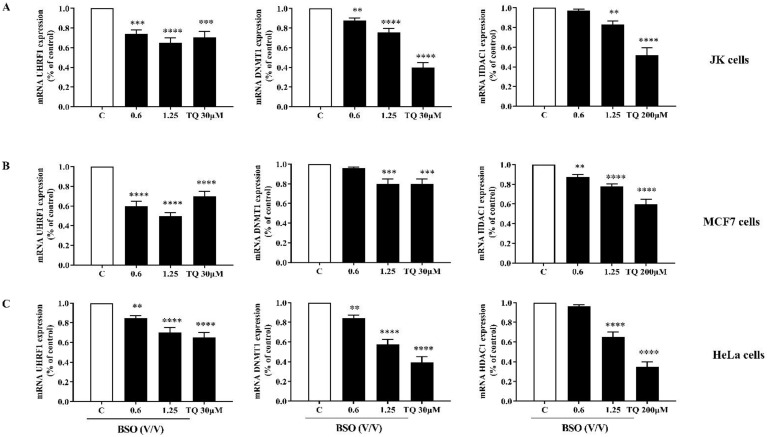
Effect of BSO and TQ mRNA expressions of UHRF1, DNMT1 and HDAC1 in cancer cells. Jurkat cells (**A**), MCF7 cells (**B**) and HeLa cells (**C**) were exposed to either BSO or TQ for 24 h. The histograms show the quantification data of mRNA expressions of UHRF1, DNMT1 and HDAC1 as assessed by RT-qPCR (**A**). Values are shown as means ± SD (n = 3); *** p* < 0.01, **** p* < 0.001 and ***** p* < 0.0001 versus respective control.

**Figure 7 molecules-27-02779-f007:**
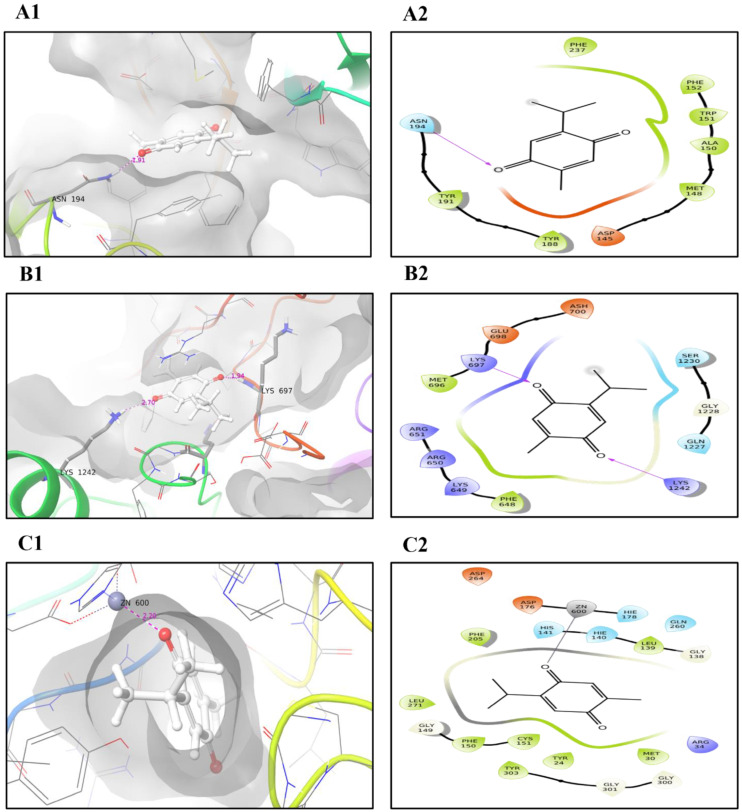
The 3D and 2D interactions of thymoquinone with proteins during the induced fit docking process where hydrogen bonds are shown by dashed lines on left and violet arrows on right; (**A1**,**A2**) present the 3D and 2D interactions, respectively, of the compounds and UHRF1 (**B1**,**B2**) show 3D and 2D interactions, respectively, of the compounds and DNMT1 and (**C1**,**C2**) show the 3D and 2D interactions, respectively, of the compounds and HDAC1. The zinc ion is shown as a gray ball in 3D figure (**C1**).

**Figure 8 molecules-27-02779-f008:**
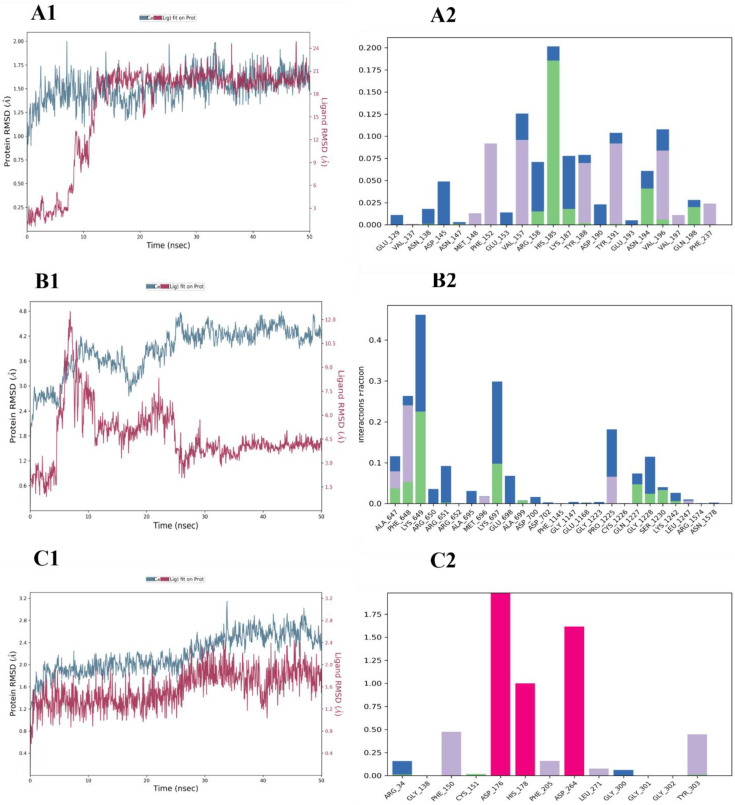
Molecular dynamic simulation of root-mean-square deviations (RMSDs) of ligand backbones are shown in red and protein in blue. On the right, the histograms show the percentage of interacting residues during 50 ns MD simulation, where the columns in each histogram indicate H-bonds (green), ionic bonds (red), water bridges (blue) and hydrophobic interactions (violet). (**A1**,**A2**) show the RMSD (left) and histogram (right) of the compounds and UHRF1; (**B1**,**B2**) show the RMSD (left) and histogram (right) of DNMT1 and the compounds and (**C1**,**C2**) show the RMSD (left) and histogram (right) of the compounds and HDAC1.

**Table 1 molecules-27-02779-t001:** Molecular docking and MD simulation properties.

PDB ID	Docking Score (Å)	SASA (Å)	Rg (Å)	MolSA (Å)	PSA (Å)
5XPI	−6.5	190	2.53	183	75
3PTA	−5	100	2.54	183	78
4BKX	−8.3	20	2.50	184	74

## Data Availability

The data presented in this study are available on request from the corresponding author.

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
