# Peer review of "The Potential Role of Nigella sativa Seed Oil as Epigenetic Therapy of Cancer"

_molecules, 2022, doi:10.3390/molecules27092779_

Round 1
Reviewer 1 Report
Research manuscript #molecules-1667513 reports an already literature known epigenetic activity of thymoquinone, a well known major component of Nigella Sativa Seed Oi. Please check literature such as: Genes 2021, 12, 622 (doi: 10.3390/genes12050622), Drug Discovery Today, 2019, 24, 2315-2322 (doi: 10.1016/j.drudis.2019.09.007); Oncotarget. 2018, 9, 28599–28611 (doi: 10.18632/oncotarget.25583). No significant additional contribution found in this manuscript to already known previous literature. It just repeats what is already known. I do not recommend publication of this manuscript.
Author Response
Answers to reviewer 1:
The authors would like to thank reviewer 1 for his (her) thoughtful comments and efforts towards improving our manuscript. The authors agree with all of them.
Reviewer #1:
Research manuscript #molecules-1667513 reports an already literature known epigenetic activity of thymoquinone, a well known major component of Nigella Sativa Seed Oi. Please check literature such as: Genes 2021, 12, 622 (doi: 10.3390/genes12050622), Drug Discovery Today, 2019, 24, 2315-2322 (doi: 10.1016/j.drudis.2019.09.007); Oncotarget. 2018, 9, 28599–28611 (doi: 10.18632/oncotarget.25583). No significant additional contribution found in this manuscript to already known previous literature. It just repeats what is already known. I do not recommend publication of this manuscript.
Author’s response:
In our previous studies, we evaluated the effects of TQ on the expression of UHRF1 and its partners DNMT1 and HDAC1 (Alhosin M et al;.Biochem Pharmacol. 2010. Alhosin M et al;.Invest New Drugs. 2012. Ibrahim A et al;., Oncotarget. 2018. lhosin M et al; applied Sciences 2021). The present study was designed to evaluate the inhibitory effects of the extracted black seed oil on the expression of the trimeric complex UHRF1/DNMT1/ HDAC1 and the related events in several cancer cells. Thus, a pure TQ was used as a positive control, as it is the main compound active compound in BSO and it induces apoptosis in several cancer cells including the MCF-7, HeLa and Jurkat cells through inhibiting the expression of the trimeric complex UHRF1/DNMT1/ HDAC1. To avoid any confusion and misunderstanding for readers and to highlight the novelty of the present study, we edited the last paragraph in the introduction:
The aim of the present study was to extract BSO from black seeds sourced from the local market of Al-Qassim, Saudi Arabia, to determine its content of TQ and to investigate inhibitory effects of extracted BSO on the expression of UHRF1, DNMT1 and HDAC1 and the related events in several cancer cells.
Reviewer 2 Report
Here is my suggestion:
- Why choose 0.6% and 1.25% concentrations? In fact, he ability to kill cancer cells by BSO is insufficient (Figure 3). Although there are significant differences, the main reason is that the statistical method chooses SEM. If mean±SD analysis is used, it may not be effective. Importantly, showing IC50 is important so that readers can compare with known anticancer components.
- To illustrate the cancer therapy potential by BSO, the toxicity to normal cells must also be assessed.
- In my opinion, there is no value in showing Figure 5. I suggest using some methods to detect in situ apoptosis which is more meaningful.
- “HeLa” or “Hela”, please comfirm and unify.
- BSO expressed in v/v, while compound expressed in microM, how to compare.
- The descriptions of Figures 4 and 5 have been reversed.
- Results in Figure 6, the effect is also not significant, so analysis of protein expression is required.
Please check the legends of Figure 6, is “(A) Cell viability ……” redundant?

Author Response
Answers to reviewer 2:
The authors would like to thank reviewer 2 for his (her) thoughtful comments and efforts towards improving our manuscript. The authors agree with all of them.
Reviewer 2:
Here is my suggestion:
Comment 1. Why choose 0.6% and 1.25% concentrations? In fact, he ability to kill cancer cells by BSO is insufficient (Figure 3). Although there are significant differences, the main reason is that the statistical method chooses SEM. If mean±SD analysis is used, it may not be effective. Importantly, showing IC50 is important so that readers can compare with known anticancer components.
Author’s response:
For Comment 1.1: BSO significantly decreased cancer cell proliferation starting from 0.6%. Thus, 0.6% and 1.25% concentrations were used in the study.
For Comment 1.2: Regarding the statistical analysis, there was orthographic error. Indeed, all the statistical analysis was performed using mean±SD. We thank the reviewer to bring our attention for this point. We corrected it in the revised version.
For Comment 1.3: Since the BSO compositions are different according to geographical origin of the plants, IC50 will also be different according. Indeed, the BSO content of TQ has been shown to vary from as low as 0.01 mg/g to 13.30 mg/g (we added this information in the revised version). However, in our experimental conditions, calculated concentrations of BSO-induced half-maximal effects on cell proliferation were approximately 5±0.6% for 24h of treatment (data not shown).
Comment 2. To illustrate the cancer therapy potential by BSO, the toxicity to normal cells must also be assessed.
Author’s response: Different cell toxicity studies have reported that N. sativa oil is safe for human use and consumption (Ahmad A et al., Asian Pacific journal of tropical biomedicine 2013, Huseini HF, et al., Planta medica 2016, Khonche A, et al., Journal of ethnopharmacology 2019). All these references have been cited in the present article.
Comment 3. In my opinion, there is no value in showing Figure 5. I suggest using some methods to detect in situ apoptosis which is more meaningful.
Author’s response: apoptosis rate was analyzed in the present study using the Annexin V Binding Guava Nexin® Assay by capillary cytometry (Figure 4). To support these findings, we studied the morphological changes of cancer cells in response to BSO treatment (Figure 5). Compared to untreated cells, BSO treatment resulted in Jurkat cell shrinking (Figure 5A) and led to an increase in detached MCF7 cells (Figure 5B) and HeLa cells (Figure 5C) which are an indication of typical appearance of apoptotic cells.
Comment 4. “HeLa” or “Hela”, please comfirm and unify.
Author’s response: the cell line is HeLa. Hela is removed and replaced by HeLa in the revised version.
Comment 5. BSO expressed in v/v, while compound expressed in microM, how to compare.
Author’s response: As recommended in several works, plant extracts or oils including BSO could be used at concentrations v/v (L Ait Mbarek et al, Braz J Med Biol Res. 2007 , Klaudia Ciesielska-Figlon et al.Hum Immunol. 2021). TQ was expressed in microM as it is a pure compound with known molecular weight.
Comment 6. The descriptions of Figures 4 and 5 have been reversed.
Author’s response: The legends of both figures 4 and 5 have been corrected in the revised version.
Comment 7. Results in Figure 6, the effect is also not significant, so analysis of protein expression is required.
Author’s response: The present study showed that BSO treatment induced a significant decrease in mRNA expression of UHRF1, DNMT1 and HDAC1. Since the present work showed that TQ was the major volatile compound in the extracted BSO, this later is also expected to decrease the expression of the target genes at protein level. This hypothesis is supported by our previous studies showing that TQ decreases the expression of UHRF1 protein as well as its partners DNMT1 and HDAC1 in several cancer cells (Alhosin M et al;.Biochem Pharmacol. 2010. Alhosin M et al;.Invest New Drugs. 2012. Ibrahim A et al;., Oncotarget. 2018. lhosin M et al; applied Sciences 2021).
Comment 8. Please check the legends of Figure 6, is “(A) Cell viability ……” redundant?
Author’s response: the legends of Figure 6 has been corrected in the revised version by deleting: Cell viability rate was assessed by WST-1 assay (B).
Reviewer 3 Report
Regarding the manuscript entitled "The Potential Role of Nigella Sativa Seed Oil as Epigenetic Therapy of Cancer" I have some comments and questions in order to improve its quality, please find them below:
Abstract
- please make the plants' names italic throughout the manuscript
- I would recommend replacing "Black seed oil" with its scientific name in the Keywords
Introduction
- the ethnomedicinal uses of the seeds can be mentioned, specifically in Saudi Arabia
- L55,56: authors "Black seed oil (BSO), also known as Nigella sativa oil, is a an important source of components with a wide range of biological activities including anticancer effects", which type of cancer, it has shown potency against which cancer cell lines. is there any reports on the same cells selected in this study? if yes please describe the novelty
- please describe the reason for choosing "MCF-7, HeLa and Jurkat cells"
- please read carefully once more in case of the abbreviated forms while they should be defined as the first use, e.g. "SRA" in L43
Results
- Authors: "Several reports have shown that N. sativa seeds are rich in volatile constituents, but the compositions are different according to geographical origin of the plants", how difference are? please bring a percentage range
- have you applied any drugs as positive control to compare the cytotoxicity and pro-apoptotic potencies?
Discussion
L322: please revise "Nigella Sativa" to "Nigella sativa", also "Ranunculaceae2 should not be italic, check the whole text
L333-335: please describe against which cancer cells has shown cell proliferation and apoptosis effects
L375: please allocate a distinct section for "Conclusion"
References
- all must be styled according to the journal's guidelines
Author Response
Answers to reviewer 3:
The authors would like to thank reviewer 3 for his (her) thoughtful comments and efforts towards improving our manuscript. The authors agree with all of them.
Reviewer 3:
Regarding the manuscript entitled "The Potential Role of Nigella Sativa Seed Oil as Epigenetic Therapy of Cancer" I have some comments and questions in order to improve its quality, please find them below:
Abstract
Comment - please make the plants' names italic throughout the manuscript
Author’s response: The name of Nigella Sativa is italic in the revised version
Comment - I would recommend replacing "Black seed oil" with its scientific name in the Keywords
Author’s response: As requested by the reviewer, Black seed oil is replaced by Nigella Sativa oil in the Keywords in the revised version.
Introduction
Comment - the ethnomedicinal uses of the seeds can be mentioned, specifically in Saudi Arabia
Author’s response: As requested by the reviewer, we added in the introduction in the revised version a new paragraph for this statement. The new paragraph is: Beside to its anticancer effects against many types of tumors, black seeds and its oil have remedial usage for other various diseases and disorders such as diabetes, cardiovascular, hypertension, infection, inflammation, and eczema [24, 25].
Comment - L55,56: authors "Black seed oil (BSO), also known as Nigella sativa oil, is an important source of components with a wide range of biological activities including anticancer effects", which type of cancer, it has shown potency against which cancer cell lines. is there any reports on the same cells selected in this study? if yes please describe the novelty
Author’s response: Black seed oil (BSO) has been shown to exert cytotoxic effects against several cancer cells including colon cancer, lung cancer and human breast cancer cells (we added the name of these cells in the revised version). Only one previous study has evaluated the effect of BSO on MCF7 but without showing the signaling pathway which could be implicated. The novelty of our study is that we showed the effect of BSO on the expression of the epigenetic players UHRF1, DNMT1 and HDAC1 which are known to be overexpressed in several cancer cells.
Comment - please describe the reason for choosing "MCF-7, HeLa and Jurkat cells"
Authors' response:
In the present study, we used the human T lymphocyte Jurkat cell line and human MCF-7, HeLa for two raisons. First, UHRF1, which is one of the main targets in the present study is highly expressed in all cancer cell lines (this statement is supported by several references in our article). Second, in our study, we used MCF-7, HeLa, as models of solid tumors, and Jurkat cells, as a hematological tumor model to investigate the inhibitory effects of BSO in these cancer cells in order to support the potential of BSO as anticancer therapy for both blood malignancies and solid tumors.
Comment - please read carefully once more in case of the abbreviated forms while they should be defined as the first use, e.g. "SRA" in L43
Authors' response: The abbreviation for SRA domain: SET and RING-associated has been added in the revised version
Results
Comment - Authors: "Several reports have shown that N. sativa seeds are rich in volatile constituents, but the compositions are different according to geographical origin of the plants", how difference are? please bring a percentage range
Author’s response: BSO content of TQ has been shown to vary from as low as 0.01 mg/g to 13.30 mg/g. This information has been added in the revised version and supported with the reference number 47.
Comment: have you applied any drugs as positive control to compare the cytotoxicity and pro-apoptotic potencies?
Author’s response: The present study was designed to evaluate the inhibitory effects of the extracted black seed oil compared to TQ on the expression of the trimeric complex UHRF1/DNMT1/ HDAC1 and the related events in several cancer cells. A pure TQ was used as a positive control in the present study, as it is the main compound active compound in BSO and it induces apoptosis in several cancer cells including the MCF-7, HeLa and Jurkat cells through inhibiting the expression of the trimeric complex UHRF1/DNMT1/ HDAC1.
Discussion
Comment . L322: please revise "Nigella Sativa" to "Nigella sativa", also "Ranunculaceae 2 should not be italic, check the whole text
Authors' response: The name Nigella sativa is written in italic and the Ranunculaceae is in non-italic in the revised version
Comment . L333-335: please describe against which cancer cells has shown cell proliferation and apoptosis effects
Authors' response: Several in vitro and in vivo studies have shown that N. sativa volatile oil exerts anti-cancer activity on many cancer cells including human gastric cancer cells, colon cells and human breast cancer cells through targeting several signaling pathways especially those involved in cell proliferation and apoptosis. The name of these cells has been added in the revised version.
Comment . L375: please allocate a distinct section for "Conclusion"
Authors' response: As requested by the reviewer, a separated section for conclusion has been added in in the revised version.
References
Comment - all must be styled according to the journal's guidelines
Authors' response: all the references are in the style of the MDPI.
Round 2
Reviewer 2 Report
- Figure 5/C "Hela" is not changed.
Reviewer 3 Report
Authors have revised the suggested and requested revisions, the present form can be considered for further publication procedure.